evolution/genetics/health and disease and epidemiology

mitochondrion, heterosis, overdominance, CNV, qPCR

**Author for correspondence:**
Ulrich Knief
e-mail: knief@biologie.uni-muenchen.de

# A sex chromosome inversion is associated with copy number variation of mitochondrial DNA in zebra finch sperm

Ulrich Knief[1], Wolfgang Forstmeier[2], Bart Kempenaers[2] and Jochen B. W. Wolf[1]

[1]Division of Evolutionary Biology, Faculty of Biology, Ludwig Maximilian University of Munich, Planegg-Martinsried 82152, Germany
[2]Department of Behavioural Ecology and Evolutionary Genetics, Max Planck Institute for Ornithology, Seewiesen 82319, Germany

UK, 0000-0001-6959-3033; WF, 0000-0002-5984-8925; BK, 0000-0002-7505-5458; JBWW, 0000-0002-2958-5183

The propulsion of sperm cells via movement of the flagellum is of vital importance for successful fertilization. While the exact mechanism of energy production for this movement varies between species, in avian species energy is thought to come predominantly from the mitochondria located in the sperm midpiece. Larger midpieces may contain more mitochondria, which should enhance the energetic capacity and possibly promote mobility. Due to an inversion polymorphism on their sex chromosome *TguZ*, zebra finches (*Taeniopygia guttata castanotis*) exhibit large within-species variation in sperm midpiece length, and those sperm with the longest midpieces swim the fastest. Here, we test through quantitative real-time PCR in zebra finch ejaculates whether the inversion genotype has an effect on the copy number of mitochondrial DNA (mtDNA). We find that zebra finches carrying the derived allele (correlated with longer sperm midpieces) have more copies of the mtDNA in their ejaculates than those homozygous for the ancestral allele (shorter midpieces). We suggest downstream effects of mtDNA copy number variation on the rate of adenosine triphosphate production, which in turn may influence sperm swimming speed and fertilization success. Central components of gamete energy metabolism may thus be the proximate cause for a fitness-relevant genetic polymorphism, stabilizing a megabase-scale inversion at an intermediate allele frequency in the wild.

# 1. Introduction

Sperm morphology can have direct consequences for sperm motility and male reproductive success [1–5]. Thus, sperm competition and cryptic female choice should exert strong directional selection towards an optimal sperm design, presumably enhancing motility or energetic capacity [4,6,7].

A typical sperm cell moves through the activity of its flagellum [8], which consists of the midpiece and the tail [9]. It derives the energy for its movement either through anaerobic glycolysis in the tail or through oxidative phosphorylation (OXPHOS) in the mitochondria of the midpiece [8]. Taxonomic groups vary in their use of these two energetic pathways [8], and intraspecific variation seems to be common. Bedford & Hoskins [10] categorized mammalian sperm as either being predominantly powered by respiration (e.g. domesticated horse *Equus ferus caballus*; [11]), by both OXPHOS and glycolysis (e.g. boar *Sus scrofa*; [12,13]), or by glycolysis without the need for respiration (e.g. human *Homo sapiens*; [14]). Enlarged sperm tails and sperm midpieces have both been predicted to increase sperm velocity [15,16], because longer tails produce more thrust and larger midpieces may contain more mitochondria and hence produce more energy [15]. Indeed, a positive relationship between midpiece size and sperm velocity has been found across multiple taxonomic groups [4,5,17,18]. Yet, a negative relationship between mitochondrial DNA (mtDNA) copy number and sperm velocity has been observed across mammalian species, which may be due to increased reactive oxygen species (ROS) production [11,13,19–24].

Avian species use both OXPHOS and glycolysis for energy production in sperm cells (studied in Galliformes [25,26]). However, in chicken (*Gallus gallus*), OXPHOS alone supports normal sperm motility [27], suggesting respiration in mitochondria as the main energy source [6,27,28]. In birds, the total number of mitochondria per sperm cell varies from less than 20 to more than 350 across species [9]. In oscine Passerines (songbirds), the largest phylogenetic group of birds, all mitochondria fuse into a helical strand that winds along the midpiece (gyres) during spermatogenesis [29–31]. The number of mitochondria contributing to this mitochondrial syncytium is basically unknown. Because sperm midpiece length correlates with adenosine triphosphate (ATP) content in whole ejaculates across songbirds, it has been hypothesized that a larger number of mitochondria contributes to the syncytium in longer midpieces [28]. Interestingly, ROS production is lower in birds than in mammals [32], and high levels of antioxidants have been found in females' sperm storage tubules [33], which may allow avian sperm to contain more mtDNA copies than mammalian sperm cells (but see [34] for a counterexample in *Drosophila*).

Zebra finches are polymorphic for an inversion on their sex chromosome *TguZ* that spans roughly 63 megabases and 619 genes [35,36]. Kim *et al.* [37] and Knief *et al.* [2] have previously shown that this inversion has profound effects on sperm morphology, sperm velocity and fertilization success. Males that are heterozygous for the inversion have sperm with a long midpiece and a relatively short tail (figure 1a). The total flagellum length is intermediate between those of the two homozygous groups. This design seems optimal, because these sperm swim fastest and fertilize the most eggs in a competitive environment [2]. The intermediate allele frequency of the ancestral inversion haplotype in a wild Australian population (59.6%; [35]) might be explained by this heterotic effect. Taken together, this makes the zebra finch a suitable model to study the link between sperm design and swimming speed and unravel the mechanism behind the relationship between inversion genotype, midpiece length and sperm velocity. Mendonca *et al.* [6] hypothesized that longer midpieces—known to contain more gyres—result from the fusion of a larger number of mitochondria with the potential to boost energy metabolism. Two recent studies, which used sperm morphology rather than the inversion genotype as their predictor, did not fully support this hypothesis: mitochondrial volume increased with midpiece length less than expected [6] and ejaculates of males with long midpieces contained less ATP molecules than those of males with shorter midpieces [38]. However, reanalysing the data from [6] and [38] with the inversion genotype as a predictor (data taken from [37]) showed that sperm midpieces of heterozygous males tended to have larger mitochondrial volumes (though not significantly, $p = 0.21$, power $\beta = 22.2\%$) and contained similar amounts of ATP molecules in comparison with the two homozygous groups (electronic supplementary material, Methods, figure S1).

Here, we specifically test whether the inversion genotype has an effect on mitochondrial content in zebra finch ejaculates. We predict that the sperm of heterozygous males (with longer midpieces) contain more copies of mtDNA than those of homozygous individuals.

# 2. Material and methods

## 2.1. Study subjects and sperm sampling

We extracted ejaculates by abdominal massage from 36 captive zebra finches (*Taeniopygia guttata castanotis*, average ± s.d. age at sampling = 6.3 ± 0.3 years) housed in a unisex group at the Max Planck Institute for Ornithology, Seewiesen, Germany. Birds were derived from the study population 'Seewiesen-GB' (see [39]), in which the connection between inversion genotype, sperm morphology and siring success had been previously established [2,37]. Ejaculates were collected in 10 µl TNE buffer (10 mM Tris–HCl, 150 mM NaCl and 1 mM EDTA, pH 7.4), immediately frozen on dry ice and stored at −80°C until further usage.

## 2.2. Inversion genotypes

All individuals had been previously genotyped for six SNPs that unambiguously tag the three *TguZ* inversion haplotypes (named A, B and C) when using a unanimity decision rule (that is, all tag SNPs must specify the same type and missing data are not allowed; [35]). Further details on the genotyping and filtering procedure can be found elsewhere [35,40]. In a previous study, Knief *et al.* [2] have shown that haplotype A represents the ancestral state and that haplotypes B and C have similar effects on sperm morphology and sperm velocity. Thus, as in Knief *et al.* [2], we here combine them into the derived haplotype B*. After removing one individual with low sperm DNA concentration, the final sample consisted of nine individuals homozygous for the ancestral inversion haplotype (AA), 13 heterozygotes (AB*) and 13 individuals homozygous for the derived inversion type (B*B*).

## 2.3. Sperm DNA isolation

We used a custom protocol from Macherey-Nagel containing the GuEX buffer (50 mM Guanidine HCl, 10.5 mM Tris pH 8.0, 10.5 mM NaCl, 10.5 mM EDTA pH 8.0, 1 mM NaOH, pH 8.0–8.5) and Proteinase K together with the DNeasy Blood and Tissue Kit (Qiagen) for DNA isolation from whole ejaculates. The detailed protocol is provided in the electronic supplementary material.

## 2.4. Quantitative real-time PCR

To estimate the ratio of autosomal to mtDNA in zebra finch sperm, we used quantitative real-time PCR (qPCR; see also [19–21,41–43]). All qPCR reactions were set up with the Luna Universal qPCR master mix (New England BioLabs) and 150 nM of each primer in a total volume of 20 µl. We used three autosomal primer pairs that had been previously assayed for their amplification efficiency in the zebra finch [44] and designed three mitochondrial primer pairs in the *ND2*, *ND4* and *ND5* genes (electronic supplementary material, table S1). We checked that all primer pairs amplified single-copy products only using *in silico* PCR on the WUSTL3.2.4 zebra finch genome assembly [45]. We mixed DNA isolated from zebra finch liver and blood for normalization and tested the amplification efficiency of all primer pairs through a six- and seven-step $\log_{10}$ serial dilution of this standard (0.125–10 ng DNA for autosomal and 0.008–1 ng DNA for mitochondrial markers). All standard DNA samples were run in triplicates on the C1000 Touch Thermal Cycler with the optical module CFX96 Touch Real-Time PCR Detection System (Bio-Rad). An initial 5 min denaturation step at 95°C was followed by 40 cycles of 15 s at 95°C and 30 s at 62°C and a final melt curve measurement from 65°C to 95°C. We obtained raw *Cq*-values through the Bio-Rad CFX Manager software (v. 3.1.1517.0823). All primer pairs had efficiency values between 97% and 110% (electronic supplementary material, table S2).

For the actual quantification of mitochondrial and autosomal DNA (atDNA) in sperm, we followed the same protocol as above, using 1 ng of sperm DNA as our template in every qPCR reaction. To estimate the run-specific amplification efficiency, we set up a three-step dilution series covering the sperm DNA quantities, starting from 5 ng and 1 ng standard DNA template for the autosomal and mitochondrial primer pairs, respectively. We attempted to run each sample in triplicate for one autosomal and one mitochondrial marker in every qPCR run, along with the dilution series of the standard DNA. In total, we amplified every sample with seven to eight combinations of the autosomal and mitochondrial primer pairs.

We excluded seven qPCR reactions of two individuals that were outliers in terms of their *Cq*-values (more than four standard deviations from the mean *Cq*-value of that run). These were most likely

technical artefacts because the same samples were amplified normally with the same primer pairs in different reactions. The decision to remove extreme outliers was taken blind to the outcome of the study. After this removal, we kept the data from 1494 PCR reactions from 24 qPCR runs. In this final dataset, each of 35 individuals was genotyped within each run with each of the two primer pairs on average 3 times (range 1–6 times) and in total 7.3 times (range 1–12) with every primer pair ($N = 6$ primer pairs).

## 2.5. Statistical analyses

We used mixed-effects linear models for analysing the qPCR results as suggested by Steibel et al. [46] and Matz et al. [47]. Mixed-effects linear models and the commonly used $\Delta Cq$ method differ only in the fact that mixed-effects linear models are more powerful and flexible as they allow controlling for additional sources of variation [46,48].

For each qPCR run $i$ and each primer pair $p$, we estimated the efficiency ($E_{ip}$) by fitting a linear regression model of $Cq$-values on the $\log_{10}$ DNA concentrations of the standard amplification curve. We took the slope $\beta_{ip}$ of this regression to calculate the efficiency as

$$E_{ip} = 10^{(-1/\beta_{ip})}.$$

Ideally, exponential amplification leads to a doubling of the number of DNA molecules in every cycle of the PCR, which translates into an efficiency value of 2. This value can be converted into a percentage ($EP_{ip}$) by calculating

$$EP_{ip} = (E_{ip} - 1) \times 100.$$

We used two different approaches for correcting the raw $Cq$-values ($Cq_{sip}$) of each sample ($s$) for the amplification efficiency of the primer pair ($p$) used in each qPCR run ($i$). First, we calculated the relative abundance ($R_{sip}$) as described in Steibel et al. [46]:

$$R_{sip} = Cq_{sip} \times \log_2(E_{ip}).$$

Alternatively, we transformed the raw $Cq$-values into molecule counts ($M_{sip}$) as described in Matz et al. [47]:

$$M_{sip} = E_{ip}^{Cq1 - Cq_{sip}},$$

where $Cq1$ is the $Cq$-value when using a single DNA molecule as PCR template. It was either set to 37 (see [47]) or estimated from the standard curve of each primer pair of every qPCR run. Because there was no qualitative difference, we present results in which we used empirical $Cq1$-values ($Cq1_{ip}$).

We then fitted mixed-effects linear models with efficiency-corrected $Cq$-values ($R_{sip}$ or $M_{sip}$) as the dependent variables. To test our hypothesis that sperm of males with genotype AB* (longer midpieces) contain more copies of the mtDNA, we fitted the interaction between the two independent variables 'chromosome type' (that is autosome versus mitochondrion, one degree of freedom) and 'inversion genotype' (two degrees of freedom, i.e. genotype AA with short, genotype B*B* with intermediate and genotype AB* with long sperm midpieces). In both models, we corrected for individual ID (35 levels), primer pair (6 levels) and qPCR run (24 levels) by fitting them as random effects. Using $R_{sip}$ as the dependent variable, we fitted a model with a Gaussian error structure and all of the above fixed and random effects. We further analysed $M_{sip}$ by fitting a generalized linear mixed-effects model with a Poisson error structure. To account for overdispersion [49], we added an observation-level random effect (OLRE; [50]). The primer pair did not explain any variation in $M_{sip}$ and was dropped from the final model.

We also estimated the difference between autosomal and mtDNA content with the familiar $\Delta Cq$-method [51] which assumes efficiency values of 2 (=100%) for every marker and run. To this end, we first calculated the mean $Cq$-value for every sample and marker within each run, then subtracted the mean mitochondrial $Cq$-value from the mean autosomal $Cq$-value of every sample within each run and fitted a linear regression model with 'inversion genotype' as the sole predictor. Because $Cq$-values were on the $\log_2$-scale, we used them as the exponent in 2 to the power of $x$ for back-transformation. Effects were in the same direction as those obtained with the mixed-effects model using molecule counts ($M_{sip}$) as the dependent variable and fitting 'inversion genotype' was also highly significant ($p = 0.006$; electronic supplementary material, figure S2). Because mixed-effects models are more flexible and allow controlling for additional random variation [46], we present results of the mixed-effects models only.

All analyses were conducted using R (v. 3.4.3; [52]). Linear mixed-effects and generalized linear mixed-effects models were fitted with the lmer() and glmer() function of the lme4 package (v. 1.1-21;

[53]), respectively. Contrasts and $p$-values were estimated with the lmerTest package (v. 3.1-0; [54]). Standard errors of ratios of fixed effects were derived through bootstrapping using the bootMer() function with 1000 resampled datasets. Some models failed to converge during the bootstrapping procedure, but the bootstrapped fixed effect estimates and their standard errors were equivalent to those obtained through lmer() and glmer() up to the second decimal. Raw data and analyses scripts are available through the Open Science Framework (doi:10.17605/osf.io/rh4qp).

# 3. Results

As expected, zebra finch ejaculates contained significantly more mitochondrial (mtDNA) than autosomal DNA (atDNA) copies ($p = 3 \times 10^{-4}$ using relative abundance [$R_{sip}$] and $p < 2 \times 10^{-16}$ using molecule counts [$M_{sip}$]). Excluding the interaction between chromosome type and inversion genotype from the statistical model and using either the relative abundance ($R_{sip}$) or the molecule count ($M_{sip}$) data, ejaculates contained $10.5 \pm 1.5$ (expressed as a ratio ± s.e.) and $12.9 \pm 1.0$ times more copies of mitochondrial than autosomal DNA.

The amount of mtDNA relative to the amount of atDNA differed between males, depending on their *TguZ* inversion genotype. Males with genotype AA (producing sperm with short midpieces) had less mtDNA in their ejaculates than those with genotypes B*B* and AB* (producing sperm with intermediate or long midpieces, respectively; interaction $p = 3 \times 10^{-15}$ using relative abundance [$R_{sip}$] and $p = 1 \times 10^{-7}$ using molecule counts [$M_{sip}$], figure 1*b*,*c*). In the relative abundance data ($R_{sip}$), the amount of mtDNA seemed to increase linearly with each copy of the derived inversion type B* (ratio ± s.e. of mtDNA to atDNA, AA: $5.9 \pm 1.5$, AB*: $10.8 \pm 1.5$, B*B*: $14.9 \pm 1.5$; figure 1*d* and table 1). By contrast, mtDNA molecule counts ($M_{sip}$) suggested males with genotype AB* (producing sperm with the longest midpieces) had the highest mtDNA to atDNA ratio (ratio ± s.e. of mtDNA to atDNA, AA: $11.3 \pm 1.1$, AB*: $16.9 \pm 1.1$, B*B*: $11.1 \pm 1.1$; figure 1*d* and table 1).

# 4. Discussion

Here, we have shown that zebra finch ejaculates contain about 10–13 times more copies of mitochondrial than autosomal DNA. The magnitude of this difference was significantly affected by the inversion genotype of the male, such that heterozygous males have more mtDNA copies than those homozygous for the ancestral and—depending on the analyses—also the derived inversion haplotype. Since heterozygous males have the longest sperm midpieces [2,37] this difference is consistent with the hypothesized relationship between sperm morphology and energy metabolism. Although we did not directly measure sperm midpiece length in the sampled ejaculates, the inversion genotype is a reliable predictor of sperm midpiece length in this zebra finch population. The inversion genotype explains some 41.9% of the variation in midpiece length and males heterozygous for the inversion have on average 26.6% and 11.2% longer midpieces than males homozygous for the ancestral and derived genotypes, respectively [2]. It should be noted that ejaculates usually also contain small amounts of non-sperm cells [55], such that the inversion genotype could in principle also affect the cellular composition of ejaculates. However, it would require large mtDNA copy numbers in these few cells to significantly bias our estimated ratios of mitochondrial to autosomal DNA upwards.

In humans, a normal sperm cell contains about 15 mitochondria [10] and across mammalian tissues, there seems to be a consistent number of 1–3 mtDNA copies per mitochondrion ([56,57]; but [58] estimate 1–10 copies). Using qPCR, the average mtDNA copy number for a single human sperm cell is 11.13 (range = 0.2–344.9, $N = 795$ males), which translates roughly into one mtDNA copy per mitochondrion in every sperm cell [19–21,41–43]. Assuming the same to be true in passerines, then about 10–13 mitochondria could contribute to the mitochondrial syncytium of a single sperm midpiece in zebra finches, ranking lowest in comparison with other avian species (15–350 mitochondria per sperm cell; [9]). In oscine songbirds, the mitochondrial syncytium winds along the midpiece (gyres) and a single gyre in zebra finches is 3.783 µm long [6]. Taking the average midpiece lengths for the three inversion genotypes [2], short midpieces have 6.55 gyres (genotype AA), intermediate 7.46 (genotype B*B*) and long midpieces 8.29 (genotype AB*, figure 1*a*). Using the average of the relative abundance and molecule count estimates, these morphological measurements correlate well with the predicted number of 8.6 (genotype AA), 13.0 (genotype B*B*) and 13.9 (genotype AB*) mitochondria. This would mean that every gyre is stably composed of roughly 1.6 (AA: 1.3, B*B*: 1.7, AB*: 1.7) mitochondria regardless of genotype. Taking estimates from Mendonca *et al.* [6], individuals with

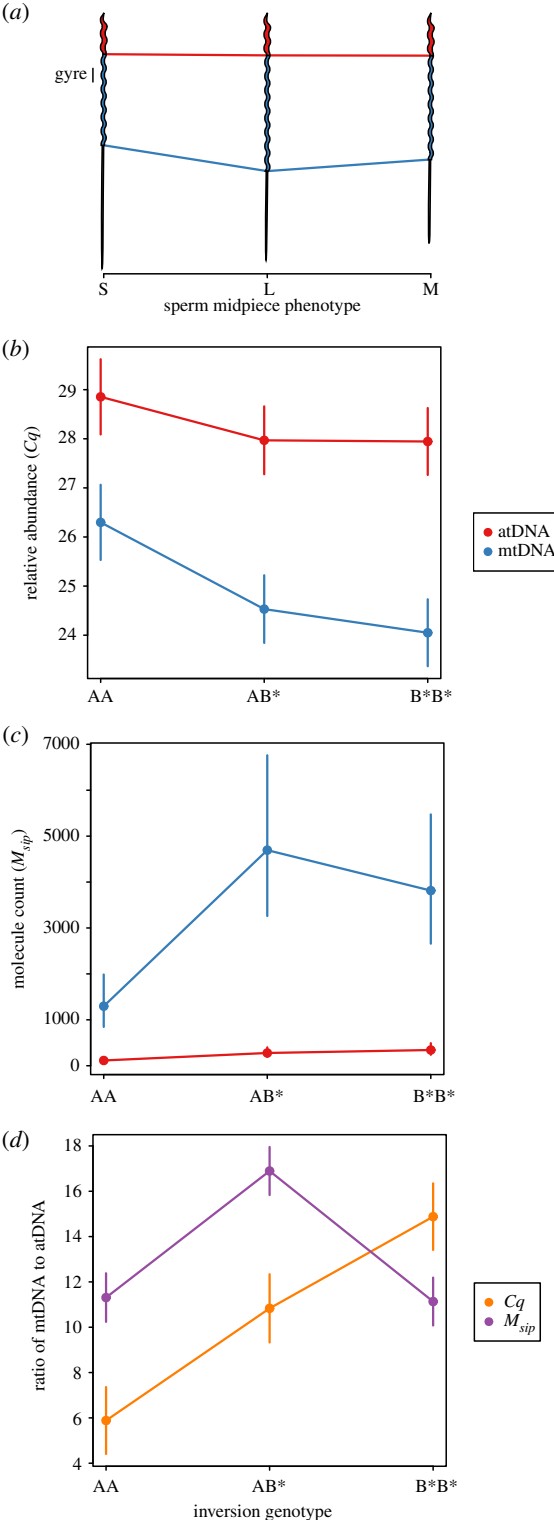

**Figure 1.** Effects of the chromosome *TguZ* inversion genotype on sperm phenotype (schematic) and on the amount (mean ± s.e.) of autosomal (nucleus, red) and mitochondrial (midpiece, blue) DNA in zebra finch ejaculates. AA and B*B* individuals are homozygous for the ancestral and derived inversion haplotypes, respectively. AB* individuals are heterozygous and have the longest midpieces. (*a*) Inversion genotype explained 41.9% of the variance in sperm midpiece length [2]. A single gyre is indicated. On the *x*-axis S, L and M denote short, long and medium midpiece lengths, which correspond to the inversion genotypes AA, AB* and B*B*, respectively. Sperm sketches were drawn to scale, with 'S'-sperm having an average length of 69.48 μm. (*b*–*d*) Effects of the inversion on DNA content were either measured as relative abundance ($R_{sip}$) *Cq*-values (*b*) or converted to molecule counts $M_{sip}$ (*c*) and were transformed into a ratio of mitochondrial to autosomal DNA (*d*, orange for $R_{sip}$ *Cq*-values and purple for $M_{sip}$). Because a lower relative abundance *Cq*-value indicates more template DNA molecules, mirroring (*b*) on the *x*-axis should result in (*c*).

**Table 1.** Estimates from the full mixed-effects models using either relative abundance ($R_{sip}$) or molecule counts ($M_{sip}$) as the response variable. Because the molecule counts model had a Poisson error structure with a log-link function, all parameter estimates are on the log-scale. There was no residual variance component estimated and the observation-level random effect ($V_{OLRE}$) represents the residual.

| response variable | parameter | estimate | s.e. | z-value | p-value |
|---|---|---|---|---|---|
| $R_{sip}$ | intercept | 28.85 | 0.77 | | |
| $N_{obs}$[a] = 1494 | chromosome type (mitochondria) | −2.56 | 0.56 | −4.55 | 0.009 |
| $N_{males}$[a] = 35 | TguZ genotype (B*B*) | −0.91 | 0.84 | −1.08 | 0.29 |
| $N_{runs}$[a] = 24 | TguZ genotype (AB*) | −0.89 | 0.85 | −1.04 | 0.30 |
| $N_{marker}$[a] = 6 | mitochondria × B*B* | −1.34 | 0.16 | −8.25 | $4 \times 10^{-16}$ |
| | mitochondria × AB* | −0.88 | 0.17 | −5.28 | $2 \times 10^{-7}$ |
| | $V_m$[b] | 3.03 | | | |
| | $V_r$[b] | 1.01 | | | |
| | $V_t$[b] | 0.45 | | | |
| | $V_R$[b] | 1.53 | | | |
| $M_{sip}$ | intercept | 4.74 | 0.43 | | |
| $N_{obs}$[a] = 1494 | chromosome type (mitochondria) | 2.43 | 0.07 | 36.02 | $<2 \times 10^{-16}$ |
| $N_{males}$[a] = 35 | TguZ genotype (B*B*) | 1.09 | 0.55 | 2.00 | 0.045 |
| $N_{runs}$[a] = 24 | TguZ genotype (AB*) | 0.89 | 0.55 | 1.62 | 0.11 |
| | mitochondria × B*B* | −0.016 | 0.087 | −0.18 | 0.86 |
| | mitochondria × AB* | 0.40 | 0.09 | 4.49 | $7 \times 10^{-6}$ |
| | $V_m$[b] | 1.34 | | | |
| | $V_r$[b] | 0.34 | | | |
| | $V_{OLRE}$[b] | 0.43 | | | |

[a]Sample sizes: In the upper model, $N_{obs}$ is only provided for information, but observation ID was not fitted as a random effect. In the lower model, marker ID was initially fitted as a random effect but explained no variance and was thus dropped from the final model.

[b]Variance components. m = male ID, r = run ID, t = marker ID, R = residual, OLRE = observation ID.

genotype AB* have about 25% larger sperm midpiece volumes than individuals of genotype AA ($V$ = 2.5 and 2.0 µm³ for AB* and AA, respectively). Yet, the number of mitochondria increases by 61% ($N$ = 13.9 and 8.6). This is similar to the 44% increase in midpiece length (mean length = 31.4 and 21.8 µm [6]) and the number of gyres (8.30 and 5.76 [6]) between individuals with genotypes AB* and AA, suggesting that the number of mitochondria in the syncytium is primarily determined by midpiece length rather than volume.

Yet, why should a larger number of mtDNA molecules increase the rate of ATP production, considering that only 13 proteins of the respiratory chain are encoded in the mtDNA and the remaining roughly 100 proteins in the nuclear DNA [59]? In non-sperm tissue, it has been shown that mtDNA copy number is correlated with respiratory activity [60] and sexually dimorphic gene expression [61], and transcript levels of the protein-coding mitochondrial genes in human sperm covary with mtDNA copy number [62]. Furthermore, mitochondrial ribosomes have been shown to be translationally active in sperm cells; inhibiting them reduces sperm motility [63]. Thus, a larger number of mtDNA copies could directly affect the number of mitochondrial ribosomes, either because mtDNA copy number serves as a proxy for the number of mitochondrial ribosomes or because more mtDNA copies would allow higher transcription rates of their ribosomal RNAs [64].

In contrast to our results, sperm of human and other mammalian males that harbour more mtDNA copies swim slower ([11,13,19–23,65,66], but see [67,68] for the opposite finding). This is often explained by larger quantities of ROS produced in sperm with more mitochondria, which could damage the sperm cells [24]. Mammalian and avian male reproductive physiology differs in several respects: (i) OXPHOS seems to be the main energy source for sperm motility in avian species, whereas glycolysis seems to be more relevant in mammals [27,28,69]. (ii) Mammalian sperm needs glycolysis for capacitation,

which is not required in birds [69]. (iii) With its helical shape, Passerine sperm is morphologically distinct from mammalian sperm, moves differently by spinning around its long axis with little bending of the tail [30] and thus probably follows different hydrodynamic models [70]. (iv) In birds, the females' sperm storage tubules provide an antioxidant environment that shields sperm from ROS-induced damage for prolonged periods of time (at least a week and up to several months), which is considerably longer than in mammals [33]. (v) Despite their increased energy demands, birds effectively reduce ROS production in (somatic) tissues [32]. Taken together, findings in mammals may not be directly transferable to birds.

Summarizing, this study establishes an intriguing link between naturally segregating inversion genotypes, sperm morphology and physiology, and motivates future research on energy metabolism in male gametes of non-mammalian species.

Ethics. Housing, breeding, blood and sperm sampling of the captive zebra finches are covered by the housing and breeding permit granted to W.F. (# 311.4-si, Landratsamt Starnberg, Germany).

Data accessibility. Raw qRT-PCR data and analyses scripts are accessible through the Open Science Framework (doi:10.17605/osf.io/rh4qp).

Author's contribution. U.K. and J.B.W.W. conceived of the study. U.K. performed research and analysed the data. W.F. and B.K. provided samples and intellectual input. U.K. wrote the paper with input from all authors.

Competing interests. The authors declare no competing interests.

Funding. Funding was provided by the LMU Munich (to J.B.W.W.) and the Max Planck Society (to B.K.).

Acknowledgements. We thank K. Martin for help with breeding and sperm sampling and S. Bauer, E. Bodendorfer, A. Grötsch, A. Kortner, K. Martin, P. Neubauer, F. Weigel and B. Wörle for animal care. F. Martínez-Pastor provided the TNE buffer recipe. We are further grateful to G. Kumpfmüller for laboratory work and thank J. Peñalba for the sperm illustrations. We thank two anonymous reviewers for critically reading the manuscript and suggesting substantial improvements.

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
