## [Peer Review File · Royal Society Open Science]

Review History

RSOS-210156.R0 (Original submission)

Review form: Reviewer 1

Is the manuscript scientifically sound in its present form?

No

Are the interpretations and conclusions justified by the results?

No

Is the language acceptable?

No

Do you have any ethical concerns with this paper?

No

Have you any concerns about statistical analyses in this paper?

Yes

Recommendation?

Reject

Comments to the Author(s)

In this manuscript, the authors investigate the association between mtDNA copy number and sperm haplotype for a sex chromosome inversion. The authors argue that haplotype predicts mtDNA copy number and suggest that this finding is related to sperm midpiece length and motility, however sperm morphology and motility of the samples is not presented here. The paper shows surprising findings that would conflict with most previously published reports on mtDNA copy number in sperm if the assumption that the haplotype of these sperm samples does indeed indicate morphology and motility. The paper suffers from methodical and foundational flaws as described below.

Overall, the authors do not provide a scholarly review of relevant literature to frame their study. For example, the references 1, 2 & 4 in the first paragraph of the introduction are narrow studies used to support very broad statements. I suggest the author review the literature more broadly to find stronger support for their statements. Related to this, are the following specific comments:

Line 3 – the sentence, “Presumably, the energy for this movement comes from the mitochondria...” suggests that it is not known where the energy is coming from but it is likely the mitochondria. The energetics of sperm movement is very well characterized, as stated in the introduction. Suggest the authors rephrase the abstract.

Line 21 – the “proximate connection between sperm design and motility remains poorly understood” is not true. There are a number of theoretical, empirical and comparative studies on this topic, and a plethora of reviews. I suggest the authors consider the relevant literature on this topic.

The functional significance of mtDNA copy number in sperm has been well studied and previous findings overwhelmingly conflict with this study, yet this is not discussed until the final paragraph of the discussion and then the focus is on human studies. The authors instead predict that mtDNA copy number will positively associate with sperm motility and midpiece length but this prediction is not grounded in our understanding of sperm physiology. For example, an empirical study showed that males with an experimentally induced 3-fold decrease in mtDNA copy number did not experience any decrease in fertility (Wai et al. 2010). Moreover, most studies show an inverse relationship between mtDNA copy number and sperm motility, for example in boar (eg., Guo et al. 2016), stallions (eg., Darr et al., 2017), dogs, mice, as well as a number of studies in humans. Why is the prediction that is laid out in the introduction not consistent with previous studies examining the function of mtDNA copy number in sperm? In addition, the methods used to estimate mtDNA copy number in these and other studies are rather robust and routine. The authors here chose to develop their own method without noting why they did not employ previously published methods.

Lines 51-54. It is inappropriate to say that a result was found if the statistics do not support the finding. Moreover, a non-significant result was likely found because the authors appear to use a very small number of samples (N=5 in most genotypes) with overlap distributions. The dataset does not meet appear to the requirements of test used, a likelihood ratio test (see Gudicha et al. 2016).

The basis of this authors prediction is that sperm mtDNA copy number will associate with inversion haplotype because the inversion haplotype associates with sperm midpiece length and motility. The sperm samples were on hand yet the authors failed to present data on sperm midpiece length and motility, only attempting to find an association between mtDNA copy

number and haplotype. This would be a much stronger study if the authors could show a direct association between mtDNA copy number and sperm morphology and/or performance, rather than simply binning them by haplotype and assuming their morphological and performance traits?

Review form: Reviewer 2

Is the manuscript scientifically sound in its present form?

Yes

Are the interpretations and conclusions justified by the results?

Yes

Is the language acceptable?

Yes

Do you have any ethical concerns with this paper?

No

Have you any concerns about statistical analyses in this paper?

No

Recommendation?

Accept as is

Comments to the Author(s)

This is a great study and I congratulate the authors on it, from conception through to the final manuscript. The study is effectively using some nice molecular detective work to gain greater insight into the mechanisms underlying the previously demonstrated functional difference in the sperm of two inversion karyotypes in the z chromosome of the zebra finch. In their study, they have used quantitative PCR to measure the amount of mitochondria in the sperm of different types of male (relative to autosomal sperm). The results are quite clear and consistent with what would have been expected based on the difference in the size of the midpiece of the different Z chromosome inversion karyotypes. As such this is a really nice and clear paper.

Decision letter (RSOS-210156.R0)

Dear Dr Knief

The Editors assigned to your paper RSOS-210156 "A sex chromosome inversion is associated with copy number variation of mitochondrial DNA in zebra finch sperm" have made a decision based on their reading of the paper and any comments received from reviewers.

Regrettably, in view of the reports received, the manuscript has been rejected in its current form. However, a new manuscript may be submitted which takes into consideration these comments.

We invite you to respond to the comments supplied below and prepare a resubmission of your manuscript. Below the referees' and Editors' comments (where applicable) we provide additional requirements. We provide guidance below to help you prepare your revision.

Please note that resubmitting your manuscript does not guarantee eventual acceptance, and we do not generally allow multiple rounds of revision and resubmission, so we urge you to make every effort to fully address all of the comments at this stage. If deemed necessary by the Editors, your manuscript will be sent back to one or more of the original reviewers for assessment. If the original reviewers are not available, we may invite new reviewers.

Please resubmit your revised manuscript and required files (see below) no later than 28-Sep-2021. Note: the ScholarOne system will 'lock' if resubmission is attempted on or after this deadline. If you do not think you will be able to meet this deadline, please contact the editorial office immediately.

Please note article processing charges apply to papers accepted for publication in Royal Society Open Science (<https://royalsocietypublishing.org/rsos/charges>). Charges will also apply to papers transferred to the journal from other Royal Society Publishing journals, as well as papers submitted as part of our collaboration with the Royal Society of Chemistry (<https://royalsocietypublishing.org/rsos/chemistry>). Fee waivers are available but must be requested when you submit your manuscript (<https://royalsocietypublishing.org/rsos/waivers>).

Thank you for submitting your manuscript to Royal Society Open Science and we look forward to receiving your resubmission. If you have any questions at all, please do not hesitate to get in touch.

on behalf of Dr Polly Campbell (Associate Editor) and Kevin Padian (Subject Editor)
openscience@royalsociety.org

Subject Editor Comments to Author:

Thank you for your submission. I am electing to log a "reject/resub" decision because it will give you more time to consider the reviewers' comments and revise; please don't take it as a comment on the quality of the research. Nevertheless there are some strong concerns from one reviewer that I hope you will be able to answer point by point in your resubmission. Best of luck with this.

Associate Editor Comments to Author (Dr Polly Campbell):

I agree with reviewer 2 that this study is nicely presented and that the potential insight into the molecular correlates of sperm phenotypes is valuable. However, Reviewer 1 raises a number of important concerns that need to be addressed. Framing the study in the context of prior work on sperm energetics/mtDNA copy number in taxa other than zebra finch is particularly important and will enhance the relevance of the work. I encourage the authors to add an analysis of their

qPCR data using methods from one of the studies suggested by Reviewer 1. This will also broaden the relevance of the findings. Finally, it is important to acknowledge that the relationship between mtDNA copy number and Z chromosome inversion genotype is several steps removed from sperm physiology and morphology. The fact that the studies cited in the introduction did not find a simple positive relationship between midpiece length and mitochondrial volume, and found a negative relationship between ATP and midpiece length, highlights this point.

Additional comments:

Figure 1a took some guesswork to interpret. It looks like a graph but the Y axis seems to be partially based on the number of gyres. Is this meant to show the relative length of the midpiece? I assume that S, L, M are short, long, medium but this is not indicated. I also assume that S, L, M are meant to correspond to the AA, AB* and B*B* genotypes but this is similarly unclear. Please improve the clarity of this figure.

L64-66: Please provide a citation to support this statement.

Reviewer comments to Author:

Reviewer: 1

Comments to the Author(s)

In this manuscript, the authors investigate the association between mtDNA copy number and sperm haplotype for a sex chromosome inversion. The authors argue that haplotype predicts mtDNA copy number and suggest that this finding is related to sperm midpiece length and motility, however sperm morphology and motility of the samples is not presented here. The paper shows surprising findings that would conflict with most previously published reports on mtDNA copy number in sperm if the assumption that the haplotype of these sperm samples does indeed indicate morphology and motility. The paper suffers from methodical and foundational flaws as described below.

Overall, the authors do not provide a scholarly review of relevant literature to frame their study. For example, the references 1, 2 & 4 in the first paragraph of the introduction are narrow studies used to support very broad statements. I suggest the author review the literature more broadly to find stronger support for their statements. Related to this, are the following specific comments:

Line 3 - the sentence, "Presumably, the energy for this movement comes from the mitochondria..." suggests that it is not known where the energy is coming from but it is likely the mitochondria. The energetics of sperm movement is very well characterized, as stated in the introduction. Suggest the authors rephrase the abstract.

Line 21 - the "proximate connection between sperm design and motility remains poorly understood" is not true. There are a number of theoretical, empirical and comparative studies on this topic, and a plethora of reviews. I suggest the authors consider the relevant literature on this topic.

The functional significance of mtDNA copy number in sperm has been well studied and previous findings overwhelmingly conflict with this study, yet this is not discussed until the final paragraph of the discussion and then the focus is on human studies. The authors instead predict that mtDNA copy number will positively associate with sperm motility and midpiece length but this prediction is not grounded in our understanding of sperm physiology. For example, an empirical study showed that males with an experimentally induced 3-fold decrease in mtDNA copy number did not experience any decrease in fertility (Wai et al. 2010). Moreover, most studies show an inverse relationship between mtDNA copy number and sperm motility, for example in

boar (eg., Guo et al. 2016), stallions (eg., Darr et al., 2017), dogs, mice, as well as a number of studies in humans. Why is the prediction that is laid out in the introduction not consistent with previous studies examining the function of mtDNA copy number in sperm? In addition, the methods used to estimate mtDNA copy number in these and other studies are rather robust and routine. The authors here chose to develop their own method without noting why they did not employ previously published methods.

Lines 51-54. It is inappropriate to say that a result was found if the statistics do not support the finding. Moreover, a non-significant result was likely found because the authors appear to use a very small number of samples (N=5 in most genotypes) with overlap distributions. The dataset does not meet appear to the requirements of test used, a likelihood ratio test (see Gudicha et al. 2016).

The basis of this authors prediction is that sperm mtDNA copy number will associate with inversion haplotype because the inversion haplotype associates with sperm midpiece length and motility. The sperm samples were on hand yet the authors failed to present data on sperm midpiece length and motility, only attempting to find an association between mtDNA copy number and haplotype. This would be a much stronger study if the authors could show a direct association between mtDNA copy number and sperm morphology and/or performance, rather than simply binning them by haplotype and assuming their morphological and performance traits?

Reviewer: 2

Comments to the Author(s)

This is a great study and I congratulate the authors on it, from conception through to the final manuscript. The study is effectively using some nice molecular detective work to gain greater insight into the mechanisms underlying the previously demonstrated functional difference in the sperm of two inversion karyotypes in the z chromosome of the zebra finch. In their study, they have used quantitative PCR to measure the amount of mitochondria in the sperm of different types of male (relative to autosomal sperm). The results are quite clear and consistent with what would have been expected based on the difference in the size of the midpiece of the different Z chromosome inversion karyotypes. As such this is a really nice and clear paper.

===PREPARING YOUR MANUSCRIPT===

While not essential, it will speed up the preparation of your manuscript proof if accepted if you format your references/bibliography in Vancouver style (please see

<https://royalsociety.org/journals/authors/author-guidelines/#formatting>). You should include DOIs for as many of the references as possible.

===PREPARING YOUR REVISION IN SCHOLARONE===

Author's Response to Decision Letter for (RSOS-210156.R0)

See Appendix A.

RSOS-211025.R0

Review form: Reviewer 1

Is the manuscript scientifically sound in its present form?

Yes

Are the interpretations and conclusions justified by the results?

Yes

Is the language acceptable?

Yes

Do you have any ethical concerns with this paper?

No

Have you any concerns about statistical analyses in this paper?

No

Recommendation?

Accept with minor revision (please list in comments)

Comments to the Author(s)

The resubmission of the MS is much improved and communicates an interesting study that builds on our understanding of the genetic basis of sperm morphology and energetics.

Overall, the framing of the study is improved and considered more of the relevant published literature, yet draws important distinctions between avian sperm and other groups such as mammals.

My one objection to the study is that I am uncomfortable with the idea of using “inversion genotype as a proxy for midpiece length” (i.e., Line 10-11 and elsewhere). I suggest the authors report their results directly (mtDNA copy number and inversion genotype), and then add that the inversion genotype is strongly associated with midpiece length, which suggests a positive association between mtDNA copy number and midpiece length. I appreciate the way the authors laid out the goal of the study in the introduction (Lines 73-75), and suggest sticking with this framing through the text.

The statistical analysis in the revision, and the authors’ explanation of their methods, is also improved. The analyses now seem sound and the results robust, especially since they are consistent with the more traditional method for estimating mtDNA copy number in sperm.

Lastly, I commend the authors on a thorough and interesting interpretation of the work in the discussion. Drawing clear differences between the better studied mammalian sperm and the focus of this work, finch sperm, is helpful and adds to our broader understanding of sperm biology in diverse species.

Review form: Reviewer 2

Is the manuscript scientifically sound in its present form?

Yes

Are the interpretations and conclusions justified by the results?

Yes

Is the language acceptable?

Yes

Do you have any ethical concerns with this paper?

No

Have you any concerns about statistical analyses in this paper?

No

Recommendation?

Accept as is

Comments to the Author(s)

I was previously very positive about this paper and I am happy to see that the authors have responded very positively to the issues raised by the other reviewer. Those comments were very useful and in particular identified some issues around sperm motility and function in the literature and from work on other taxa that certainly should have been covered in the introduction. The improved perspective in the revised version certainly helps to make the manuscript more rounded and insightful. Once again I feel that this is a really nice study that adds significant insight to our understanding of sperm structure and function.

Decision letter (RSOS-211025.R0)

Dear Dr Knief

On behalf of the Editors, we are pleased to inform you that your Manuscript RSOS-211025 "A sex chromosome inversion is associated with copy number variation of mitochondrial DNA in zebra finch sperm" has been accepted for publication in Royal Society Open Science subject to minor revision in accordance with the referees' reports. Please find the referees' comments along with any feedback from the Editors below my signature.

Please submit your revised manuscript and required files (see below) no later than 7 days from today's (ie 09-Jul-2021) date. Note: the ScholarOne system will 'lock' if submission of the revision is attempted 7 or more days after the deadline. If you do not think you will be able to meet this deadline please contact the editorial office immediately.

on behalf of Dr Polly Campbell (Associate Editor) and Kevin Padian (Subject Editor)
openscience@royalsociety.org

Associate Editor Comments to Author (Dr Polly Campbell):

Associate Editor

Comments to the Author:

The authors have done a very nice job in revising this manuscript; the thorough treatment of mammalian vs. avian sperm adds breadth to an already valuable study. Both reviewers are satisfied with the revision but Reviewer 1 has a remaining concern that the authors should address. And please fix the typo/missing word on L277/278.

Reviewer comments to Author:

Reviewer: 2

Comments to the Author(s)

I was previously very positive about this paper and I am happy to see that the authors have responded very positively to the issues raised by the other reviewer. Those comments were very useful and in particular identified some issues around sperm motility and function in the literature and from work on other taxa that certainly should have been covered in the introduction. The improved perspective in the revised version certainly helps to make the manuscript more rounded and insightful. Once again I feel that this is a really nice study that adds significant insight to our understanding of sperm structure and function.

Reviewer: 1

Comments to the Author(s)

The resubmission of the MS is much improved and communicates an interesting study that builds on our understanding of the genetic basis of sperm morphology and energetics.

Overall, the framing of the study is improved and considered more of the relevant published literature, yet draws important distinctions between avian sperm and other groups such as mammals.

My one objection to the study is that I am uncomfortable with the idea of using “inversion genotype as a proxy for midpiece length” (i.e., Line 10-11 and elsewhere). I suggest the authors report their results directly (mtDNA copy number and inversion genotype), and then add that the inversion genotype is strongly associated with midpiece length, which suggests a positive association between mtDNA copy number and midpiece length. I appreciate the way the authors laid out the goal of the study in the introduction (Lines 73-75), and suggest sticking with this framing through the text.

The statistical analysis in the revision, and the authors' explanation of their methods, is also improved. The analyses now seem sound and the results robust, especially since they are consistent with the more traditional method for estimating mtDNA copy number in sperm.

Lastly, I commend the authors on a thorough and interesting interpretation of the work in the discussion. Drawing clear differences between the better studied mammalian sperm and the focus of this work, finch sperm, is helpful and adds to our broader understanding of sperm biology in diverse species.

===PREPARING YOUR MANUSCRIPT===

Please ensure that you include an acknowledgements' section before your reference list/bibliography. This should acknowledge anyone who assisted with your work, but does not

qualify as an author per the guidelines at <https://royalsociety.org/journals/ethics-policies/openness/>.

===PREPARING YOUR REVISION IN SCHOLARONE===

- Ensure that your data access statement meets the requirements at <https://royalsociety.org/journals/authors/author-guidelines/#data>. You should ensure that you cite the dataset in your reference list. If you have deposited data etc in the Dryad repository, please only include the 'For publication' link at this stage. You should remove the 'For review' link.
- If you are requesting an article processing charge waiver, you must select the relevant waiver option (if requesting a discretionary waiver, the form should have been uploaded at Step 3 'File upload' above).
- If you have uploaded ESM files, please ensure you follow the guidance at <https://royalsociety.org/journals/authors/author-guidelines/#supplementary-material> to include a suitable title and informative caption. An example of appropriate titling and captioning may be found at https://figshare.com/articles/Table_S2_from_Is_there_a_trade-off_between_peak_performance_and_performance_breadth_across_temperatures_for_aerobic_scope_in_teleost_fishes_/3843624.

Author's Response to Decision Letter for (RSOS-211025.R0)

See Appendix B.

Decision letter (RSOS-211025.R1)

Dear Dr Knief,

I am pleased to inform you that your manuscript entitled "A sex chromosome inversion is associated with copy number variation of mitochondrial DNA in zebra finch sperm" is now accepted for publication in Royal Society Open Science.

You can expect to receive a proof of your article in the near future. Please contact the editorial office (openscience@royalsociety.org) and the production office (openscience_proofs@royalsociety.org) to let us know if you are likely to be away from e-mail contact -- if you are going to be away, please nominate a co-author (if available) to manage the proofing process, and ensure they are copied into your email to the journal. Due to rapid

publication and an extremely tight schedule, if comments are not received, your paper may experience a delay in publication.

on behalf of Dr Polly Campbell (Associate Editor) and Kevin Padian (Subject Editor)
openscience@royalsociety.org

Appendix A

RSOS-210156: A sex chromosome inversion is associated with copy number variation of mitochondrial DNA in zebra finch sperm

Associate Editor Comments to Author (Dr Polly Campbell):

I agree with reviewer 2 that this study is nicely presented and that the potential insight into the molecular correlates of sperm phenotypes is valuable. However, Reviewer 1 raises a number of important concerns that need to be addressed. Framing the study in the context of prior work on sperm energetics/mtDNA copy number in taxa other than zebra finch is particularly important and will enhance the relevance of the work. I encourage the authors to add an analysis of their qPCR data using methods from one of the studies suggested by Reviewer 1. This will also broaden the relevance of the findings.

--- We now cover more of the work on sperm energetics/mtDNA copy number – including other taxa – in the Introduction. However, we also stress that mammalian and avian sperm may behave differently in terms of energy production and motility (e.g. Cramer et al. 2021; Setiawan et al. 2020; lines 23–50, 270–283):

1. The relative importance of respiration (OXPHOS) versus glycolysis varies even between species of eutherian mammals (Bedford & Hoskins 1990; Cummins 2009) and also between avian species (chicken and turkey; Sexton 1974; Wishart 1982). To the best of our knowledge, there are no studies investigating the relative importance of OXPHOS and glycolysis in Passerines. However, Rowe et al. (2013) suggest that OXPHOS is the main energy source for sperm motility in Passerines.

2. Passerine sperm has a very specific design (helical head and mitochondrial structure; e.g. Birkhead et al. 2005; Vernon & Woolley 1999) and show a “spinning motility, rotating about their long axis with little lateral bending of their tail” (Cummins 2009; Vernon & Woolley 1999). Thus, passerine sperm movement may not follow the hydrodynamic models developed for mammalian sperm cells, in which longer sperm swim faster than shorter sperm cells (Cramer et al. 2021; see Kim et al. 2017; Knief et al. 2017: in zebra finch, sperm with the longest midpiece swim fastest, not the ones that are longest overall).

--- Reviewer 1 suggested using the analyses methods used in Wai et al. (2010), Darr et al. (2017) and Guo et al. (2017). They used the ΔCq -method or modifications thereof to quantify the number of mtDNA copies in sperm cells. We also used the ΔCq -method for quantification (lines 180–190) but put more trust into linear mixed-effects models (LMMs). We did not develop these LMMs but they had been suggested previously by Steibel et al. (2009) and Matz et al. (2013). These studies also showed that estimates obtained through LMMs and the familiar ΔCq -method are the same, but that LMMs are more flexible and powerful and allow controlling for additional sources of variation (Gelman & Hill 2007; Steibel et al. 2009). Thus, we stick to the results obtained with the LMMs in the main text. However, we added results obtained with the ΔCq -method to the Supplement, which essentially recapitulates the results obtained when looking at molecule counts, i.e. that heterozygous individuals have the most mtDNA copies in their ejaculates (lines 180–190, Figure S2). We also highlight in the

Methods section that LMMs are an established way of analysing qRT-PCR data (lines 139–142).

Finally, it is important to acknowledge that the relationship between mtDNA copy number and Z chromosome inversion genotype is several steps removed from sperm physiology and morphology. The fact that the studies cited in the introduction did not find a simple positive relationship between midpiece length and mitochondrial volume, and found a negative relationship between ATP and midpiece length, highlights this point.

--- We now acknowledge more clearly that the relationship between mtDNA copy number and Z chromosome inversion genotype is indirect by reformulating the Abstract (lines 12–16).

Additional comments:

Figure 1a took some guesswork to interpret. It looks like a graph but the Y axis seems to be partially based on the number of gyres. Is this meant to show the relative length of the midpiece? I assume that S, L, M are short, long, medium but this is not indicated. I also assume that S, L, M are meant to correspond to the AA, AB* and B*B* genotypes but this is similarly unclear. Please improve the clarity of this figure.

--- We thank the Editor for pointing this out. We added further explanations to the figure legend (Figure 1).

L64-66: Please provide a citation to support this statement.

--- We added citations to this statement (line 83). Please notice that the birds studied in Kim et al. (2017) and in Knief et al. (2017) were derived from the same founder individuals.

Reviewer comments to Author:

Reviewer: 1

Comments to the Author(s)

In this manuscript, the authors investigate the association between mtDNA copy number and sperm haplotype for a sex chromosome inversion. The authors argue that haplotype predicts mtDNA copy number and suggest that this finding is related to sperm midpiece length and motility, however sperm morphology and motility of the samples is not presented here. The paper shows surprising findings that would conflict with most previously published reports on mtDNA copy number in sperm if the assumption that the haplotype of these sperm samples does indeed indicate morphology and motility. The paper suffers from methodical and foundational flaws as described below.

Overall, the authors do not provide a scholarly review of relevant literature to frame their study. For example, the references 1, 2 & 4 in the first paragraph of the introduction are narrow studies used to support very broad statements. I suggest the author review the literature more broadly to find stronger support for their statements.

--- The reviewer is correct when stating that we do not cover all studies related to mtDNA copy number and sperm swimming speed / fertility. We wanted to keep the Introduction concise and focus on what is known in avian taxa. Most (all?) studies that we know of were conducted on mammalian species. As explained below, it seems quite likely that avian sperm differ from mammalian sperm in terms of energy production and hydrodynamics (Setiawan et al. 2020). Nevertheless, we now include more information on what is known from studies on mammals and highlight already in the Introduction that mtDNA copy number and sperm velocity correlate negatively in mammals (lines 23–36).

Related to this, are the following specific comments:

Line 3 – the sentence, “Presumably, the energy for this movement comes from the mitochondria...” suggests that it is not known where the energy is coming from but it is likely the mitochondria. The energetics of sperm movement is very well characterized, as stated in the introduction. Suggest the authors rephrase the abstract.

--- This may be true for some mammalian species, but not for avian sperm. The relative importance of respiration (OXPHOS) versus glycolysis varies even between and within species of eutherian mammals (Bedford & Hoskins 1990; Cummins 2009) and also between avian species (chicken and turkey; Sexton 1974; Wishart 1982). OXPHOS has been suggested to be the main energy source for sperm motility in chicken (Froman & Kirby 2005) and Passerines (Rowe et al. 2013). We now rephrased this sentence to acknowledge that the energetics of sperm movement is taxon-specific, but that in avian species OXPHOS has been suggested to be the main energy source (lines 3–5, see also lines 38–50).

Line 21 – the “proximate connection between sperm design and motility remains poorly understood” is not true. There are a number of theoretical, empirical and comparative studies on this topic, and a plethora of reviews. I suggest the authors consider the relevant literature on this topic.

--- We removed the sentence. Passerine sperm has a very specific design (helical head and mitochondrial structure; e.g. Birkhead et al. 2005; Vernon & Woolley 1999) and show a “spinning motility, rotating about their long axis with little lateral bending of their tail” (Cummins 2009; Vernon & Woolley 1999). Thus, passerine sperm movement may not follow the hydrodynamic models developed for mammalian sperm cells, in which longer sperm swim faster than shorter sperm cells (Cramer et al. 2021; see Kim et al. 2017; Knief et al. 2017: in zebra finch, sperm with the longest midpiece swim fastest, not the ones that are longest overall). We now briefly cover mammalian sperm, but we also highlight that this issue is unsolved in Passerines and that taxonomic groups may differ (lines 23–36, 270–283).

The functional significance of mtDNA copy number in sperm has been well studied and previous findings overwhelmingly conflict with this study, yet this is not discussed until the final paragraph of the discussion and then the focus is on human studies. The authors instead predict that mtDNA copy number will positively associate with sperm motility and midpiece length but this prediction is not grounded in our understanding of sperm physiology. For example, an empirical study showed that males with an experimentally induced 3-fold decrease in mtDNA copy number did not experience any decrease in fertility (Wai et al. 2010). Moreover, most studies show an inverse relationship between mtDNA copy number and sperm motility, for example in boar (eg., Guo et al. 2016), stallions (eg., Darr et al., 2017), dogs, mice, as well as a number of studies in humans. Why is the prediction that is laid out in the introduction not consistent with previous studies examining the function of mtDNA copy number in sperm?

--- We now highlight that mtDNA copy number correlates negatively with sperm velocity across mammalian species already in the Introduction (lines 34–36). We then explain why avian sperm may differ from mammalian sperm and formulate our hypothesis, which is based on the relevant avian literature (see Mendonca et al. 2018; Rowe et al. 2013).

In addition, the methods used to estimate mtDNA copy number in these and other studies are rather robust and routine. The authors here chose to develop their own method without noting why they did not employ previously published methods.

--- We did not develop these linear mixed-effects models but they had been suggested previously by Steibel et al. (2009) and Matz et al. (2013). These studies also showed that estimates obtained through LMMs and the familiar ΔCq -method are similar, but that LMMs are generally more flexible and powerful. Wai et al. (2010), Darr et al. (2017) and Guo et al. (2017) all used qRT-PCR with one or more single-copy nuclear genes and mitochondrial genes. They used the ΔCq -method (Livak & Schmittgen 2001) or modifications thereof to quantify the number of mtDNA copies in sperm cells. We also used qRT-PCR with three single-copy nuclear genes and three mitochondrial genes and the ΔCq -method for quantification (lines 180–190). As the ΔCq -method gave similar results as the LMMs (molecule counts) and because the LMMs have been shown to perform better and allow controlling for additional sources of variation (Gelman & Hill 2007; Steibel et al. 2009), we keep the results obtained with the LMMs in the main text. However, we added results from the ΔCq -method to the Supplement (lines 180–190, Figure S2). We also highlight in the Methods section that LMMs are an established way of analysing qRT-PCR data (lines 139–142).

Lines 51-54. It is inappropriate to say that a result was found if the statistics do not support the finding. Moreover, a non-significant result was likely found because the authors appear to use a very

small number of samples (N=5 in most genotypes) with overlap distributions. The dataset does not meet appear to the requirements of test used, a likelihood ratio test (see Gudicha et al. 2016).

--- We agree with the reviewer, but we would like to emphasize that this lack of power does not concern our study, but rather a published one. We analyzed these published data to clarify that the patterns found in this small sample-size study are equivocal, and might well be compatible with the – not unlikely – scenario of volume increasing with length (which was found, but non-significant). We added a power analysis (power $\beta = 22.2\%$) to clarify that a type-II error is rather likely. We feel that our cautious conclusion that “sperm midpieces of heterozygous males tended to have larger mitochondrial volumes (though not significantly)” (lines 67–70) is warranted ($P = 0.21$, power $\beta = 22.2\%$). The P -value for genotype was derived using F -tests (and the Satterthwaite approximation for the denominator degrees of freedom in LMMs) as this has been shown to perform best in terms of the type-I error rate in small samples (Luke 2017). We added this to the Supplementary Methods (lines S25–S43).

The basis of this authors prediction is that sperm mtDNA copy number will associate with inversion haplotype because the inversion haplotype associates with sperm midpiece length and motility. The sperm samples were on hand yet the authors failed to present data on sperm midpiece length and motility, only attempting to find an association between mtDNA copy number and haplotype. This would be a much stronger study if the authors could show a direct association between mtDNA copy number and sperm morphology and/or performance, rather than simply binning them by haplotype and assuming their morphological and performance traits?

--- The reviewer is correct. In a previous study, we had shown that inversion haplotype explains 41.9% of the variation in sperm midpiece length and 10.0% of the variation in sperm swimming speed in the same population of birds (Knief et al. 2017). These results suggest that the relationship between inversion haplotype, sperm midpiece length and motility holds also in the males we sampled for the current study. In addition, inversion haplotype may have even stronger effects on mtDNA content in sperm than predicted by sperm midpiece length.

In the best of all worlds, we would have made morphological measurements on the same males and the same ejaculates that we used for DNA extraction to minimize the among-individual and among-ejaculate variation. Unfortunately, (1) the birds have died and cannot be sampled again and (2) zebra finch ejaculates are small, so that it is not feasible to use one ejaculate for both DNA extraction and morphological measurements.

Reviewer: 2

Comments to the Author(s)

This is a great study and I congratulate the authors on it, from conception through to the final manuscript. The study is effectively using some nice molecular detective work to gain greater insight into the mechanisms underlying the previously demonstrated functional difference in the sperm of two inversion karyotypes in the z chromosome of the zebra finch. In their study, they have used quantitative PCR to measure the amount of mitochondria in the sperm of different types of male (relative to autosomal sperm). The results are quite clear and consistent with what would have been expected based on the difference in the size of the midpiece of the different Z chromosome inversion karyotypes. As such this is a really nice and clear paper.

--- We thank the reviewer for the positive feedback and are delighted to read that our results were convincing.

References

- Bedford JM, Hoskins DD (1990) The mammalian spermatozoon: morphology, biochemistry and physiology. In: *Marshall's physiology of reproduction: reproduction in the male* (ed. Lamming GE), pp. 379–568. Churchill Livingstone, Edinburgh, UK.
- Birkhead TR, Pellatt EJ, Brekke P, Yeates R, Castillo-Juarez H (2005) Genetic effects on sperm design in the zebra finch. *Nature* **434**, 383–387.
- Cramer ERA, Garcia-del-Rey E, Johannessen LE, Laskemoen T, Marthinsen G, Johnsen A, Lifjeld JT (2021) Longer sperm swim more slowly in the Canary islands chiffchaff. *Cells* **10**, e1358.
- Cummins J (2009) Sperm motility and energetics. In: *Sperm biology: an evolutionary perspective* (eds. Birkhead TR, Hosken DJ, Pitnick S), pp. 185–206. Elsevier, London, UK.
- Darr CR, Moraes LE, Connon RE, Love CC, Teague S, Varner DD, Meyers SA (2017) The relationship between mitochondrial DNA copy number and stallion sperm function. *Theriogenology* **94**, 94–99.
- Froman DP, Kirby JD (2005) Sperm mobility: Phenotype in roosters (*Gallus domesticus*) determined by mitochondrial function. *Biol Reprod* **72**, 562–567.
- Gelman A, Hill J (2007) *Data analysis using regression and multilevel/hierarchical models*, 1 edn. Cambridge University Press, New York.
- Guo HD, Gong YB, He B, Zhao RQ (2017) Relationships between mitochondrial DNA content, mitochondrial activity, and boar sperm motility. *Theriogenology* **87**, 276–283.
- Kim KW, Bennison C, Hemmings N, Brookes L, Hurley LL, Griffith SC, . . . Slate J (2017) A sex-linked supergene controls sperm morphology and swimming speed in a songbird. *Nat Ecol Evol* **1**, 1168–1176.
- Knief U, Forstmeier W, Pei YF, Ihle M, Wang DP, Martin K, . . . Kempnaers B (2017) A sex-chromosome inversion causes strong overdominance for sperm traits that affect siring success. *Nat Ecol Evol* **1**, 1177–1184.
- Livak KJ, Schmittgen TD (2001) Analysis of relative gene expression data using real-time quantitative PCR and the $2^{-\Delta\Delta CT}$ method. *Methods* **25**, 402–408.
- Luke SG (2017) Evaluating significance in linear mixed-effects models in R. *Behavior Research Methods* **49**, 1494–1502.
- Matz MV, Wright RM, Scott JG (2013) No control genes required: Bayesian analysis of qRT-PCR data. *Plos One* **8**, e71448.
- Mendonca T, Birkhead TR, Cadby AJ, Forstmeier W, Hemmings N (2018) A trade-off between thickness and length in the zebra finch sperm mid-piece. *P R Soc B* **285**, e20180865.
- Rowe M, Laskemoen T, Johnsen A, Lifjeld JT (2013) Evolution of sperm structure and energetics in passerine birds. *P R Soc B* **280**, e20122616.
- Setiawan R, Priyadarshana C, Tajima A, Travis AJ, Asano A (2020) Localisation and function of glucose transporter GLUT1 in chicken (*Gallus gallus domesticus*) spermatozoa: relationship between ATP production pathways and flagellar motility. *Reproduction, Fertility and Development* **32**, 697–705.
- Sexton TJ (1974) Oxidative and glycolytic activity of chicken and turkey spermatozoa. *Comparative biochemistry and physiology B: Comparative biochemistry* **48**, 59–65.
- Steibel JP, Poletto R, Coussens PM, Rosa GJ (2009) A powerful and flexible linear mixed model framework for the analysis of relative quantification RT-PCR data. *Genomics* **94**, 146–152.
- Vernon GG, Woolley DM (1999) Three-dimensional motion of avian spermatozoa. *Cell Motil Cytoskel* **42**, 149–161.
- Wai T, Ao A, Zhang XY, Cyr D, Dufort D, Shoubridge EA (2010) The role of mitochondrial DNA copy number in mammalian fertility. *Biol Reprod* **83**, 52–62.
- Wishart GJ (1982) Maintenance of ATP concentrations in and of fertilizing ability of fowl and turkey spermatozoa *in vitro*. *Journal of Reproduction and Fertility* **66**, 457–462.

Appendix B

RSOS-211025: A sex chromosome inversion is associated with copy number variation of mitochondrial DNA in zebra finch sperm

Associate Editor Comments to Author (Dr Polly Campbell):

The authors have done a very nice job in revising this manuscript; the thorough treatment of mammalian vs. avian sperm adds breadth to an already valuable study. Both reviewers are satisfied with the revision but Reviewer 1 has a remaining concern that the authors should address. And please fix the typo/missing word on L277/278.

--- We fixed the typo in lines 278–279 by deleting “sperm and”. We also addressed the concern of Reviewer 1 throughout the manuscript by formulating the relationship between inversion genotype and sperm midpiece length more cautiously.

Reviewer comments to Author:

Reviewer: 1

Comments to the Author(s)

The resubmission of the MS is much improved and communicates an interesting study that builds on our understanding of the genetic basis of sperm morphology and energetics.

Overall, the framing of the study is improved and considered more of the relevant published literature, yet draws important distinctions between avian sperm and other groups such as mammals.

--- We thank the reviewer for the insightful comments and the hints to the relevant literature. We are delighted that we improved the manuscript and that the reviewer is satisfied.

My one objection to the study is that I am uncomfortable with the idea of using "inversion genotype as a proxy for midpiece length" (i.e., Line 10-11 and elsewhere). I suggest the authors report their results directly (mtDNA copy number and inversion genotype), and then add that the inversion genotype is strongly associated with midpiece length, which suggests a positive association between mtDNA copy number and midpiece length. I appreciate the way the authors laid out the goal of the study in the introduction (Lines 73-75), and suggest sticking with this framing through the text.

--- We now report our results directly. To guide the reader, we still provide information on midpiece length in brackets (lines 10–13, 173–177, 215–218, 221–222, 256–262).

The statistical analysis in the revision, and the authors' explanation of their methods, is also improved. The analyses now seem sound and the results robust, especially since they are consistent with the more traditional method for estimating mtDNA copy number in sperm.

--- We are delighted that the statistical analyses are convincing.

Lastly, I commend the authors on a thorough and interesting interpretation of the work in the discussion. Drawing clear differences between the better studied mammalian sperm and the focus of this work, finch sperm, is helpful and adds to our broader understanding of sperm biology in diverse species.

--- We thank the reviewer for the positive feedback.

Reviewer: 2

Comments to the Author(s)

I was previously very positive about this paper and I am happy to see that the authors have responded very positively to the issues raised by the other reviewer. Those comments were very useful and in particular identified some issues around sperm motility and function in the literature and from work on other taxa that certainly should have been covered in the introduction. The improved perspective in the revised version certainly helps to make the manuscript more rounded and insightful. Once again I feel that this is a really nice study that adds significant insight to our understanding of sperm structure and function.

--- We thank the reviewer for the positive feedback and are delighted to read that the manuscript improved further.